# From Development to Place in Therapy of Lorlatinib for the Treatment of ALK and ROS1 Rearranged Non-Small Cell Lung Cancer (NSCLC)

**DOI:** 10.3390/diagnostics14010048

**Published:** 2023-12-25

**Authors:** Laura Fabbri, Alessandro Di Federico, Martina Astore, Virginia Marchiori, Agnese Rejtano, Renata Seminerio, Francesco Gelsomino, Andrea De Giglio

**Affiliations:** 1Department of Medical and Surgical Sciences, Alma Mater Studiorum University of Bologna, 40126 Bologna, Italy; laura.fabbri21@studio.unibo.it (L.F.); alessandrodifederico1@gmail.com (A.D.F.); martina.astore@studio.unibo.it (M.A.); virginia.marchiori2@studio.unibo.it (V.M.); agnese.rejtano@studio.unibo.it (A.R.); renata.seminerio@studio.unibo.it (R.S.); 2Division of Medical Oncology, IRCCS Azienda Ospedaliero-Universitaria Di Bologna, 40138 Bologna, Italy; francesco_gelsomino@aosp.bo.it

**Keywords:** lorlatinib, tyrosine kinase inhibitors, anaplastic large cell lymphoma, ROS proto-oncogene 1 (ROS1), non-small cell lung cancer, central nervous system, resistance, adverse events

## Abstract

Following the results of the CROWN phase III trial, the third-generation macrocyclic ALK inhibitor lorlatinib has been introduced as a salvage option after the failure of a first-line TKI in ALK-rearranged NSCLC, while its precise role in the therapeutic algorithm of ROS1 positive disease is still to be completely defined. The ability to overcome acquired resistance to prior generation TKIs (alectinib, brigatinib, ceritinib, and crizotinib) and the high intracranial activity in brain metastatic disease thanks to increased blood–brain barrier penetration are the reasons for the growing popularity and interest in this molecule. Nevertheless, the major vulnerability of this drug resides in a peculiar profile of related collateral events, with neurological impairment being the most conflicting and debated clinical issue. The cognitive safety concern, the susceptibility to heterogeneous resistance pathways, and the absence of a valid alternative in the second line are strongly jeopardizing a potential paradigm shift in this oncogene-addicted disease. So, when prescribing lorlatinib, clinicians must face two diametrically opposed characteristics: a great therapeutic potential without the intrinsic limitations of its precursor TKIs, a cytotoxic activity threatened by suboptimal tolerability, and the unavoidable onset of resistance mechanisms we cannot properly manage yet. In this paper, we give a critical point of view on the stepwise introduction of this promising drug into clinical practice, starting from its innovative molecular and biochemical properties to intriguing future developments, without forgetting its weaknesses.

## 1. Introduction

With an estimated 2.20 million new cases and 1.79 million deaths per year, lung cancer (LC) represents the most common cancer diagnosis worldwide and remains the leading cause of cancer-related deaths. It is the second most common cancer in men, after prostate cancer, and the second most common cancer in women, after breast cancer [1]. 

Non-small cell lung cancer (NSCLC) accounts for approximately 85% of all types of lung cancer. Despite the introduction of screening guidelines and a decrease in tobacco use, which has reduced the mortality rate in the US and Western nations in recent decades, the 5-year survival rate for patients with metastatic disease, which represents the majority of patients, is still approximately 4% [2]. 

The discovery of tumor-specific alterations in oncogenes capable of driving cancer growth and survival has revolutionized the approach to cancer treatment [3]. 

Genetic sequencing has dramatically changed the management of LC over the past decades [3].

Rapid advances in next-generation sequencing and a better understanding of cancer biology have provided an opportunity to characterize human cancer genomes, including LC, leading to the identification of several actionable driver alterations implicated in the etiopathogenesis and in the subsequent development of targeted therapies [3].

In current clinical practice, several targeted agents are routinely used to treat patients with NSCLC harboring actionable genomic alterations in several genes: activating mutations in the tyrosine kinase domain of the *EGFR* gene (occurs in 10–30% of NSCLC with the incidence increasing up to 60% in Asians) [4], *KRAS* mutations (30%), *ALK* gene rearrangements (3–7%), *MET* variants (3–4%), *BRAF* mutations (1–2%), RET alterations (1–2%), *HER2* mutations (1–2%), *ROS1* chromosomal rearrangements (1–2%) [3].

As is common knowledge, targeted therapies are more effective and manageable and have fewer side effects than standard chemotherapy [5], representing the first choice of treatment for these types of tumors. Further reinforcing the need to evaluate the mutational status of tumor cells is the confirmation of a low response to immunotherapy, especially for EGFR and KRAS mutations [6].

This review aims to explore the role of third-generation lorlatinib in the treatment of NSCLCs presenting ALK and ROS1 aberrations. 

## 2. A Brief Overview of NSCLC with ALK and ROS 1 Rearrangement

The *ALK* gene encodes a tyrosine kinase receptor and is localized on the short arm of chromosome 2 (2p23), belonging to the insulin receptor superfamily and coding for the ALK protein, involved in brain development, but is also expressed in scattered adult cells such as neurons, glial cells, testis, pituitary gland, hypothalamus, and endothelial cells. 

*ALK* rearrangements are responsible for 3–7% of non-squamous NSCLC, with the mutual exclusion of other driver mutations. The following clinicopathologic features are peculiar to ALK-rearranged patients: young age at diagnosis (a median of 50 years), female, non-smokers/light smokers, adenocarcinoma histology with distinctive morphologic patterns such as cribriform and solid signet ring, expression of thyroid transcription factor 1, a tendency to metastasize to the pleura or pericardium, and particularly to the central nervous system [7].

ALK aberrations in NSCLC were first identified in 2007 [8]. Most of the abnormalities are chromosomal rearrangements resulting in the formation of fusion genes. The most frequent partner of fusion is echinoderm microtubule-associated protein 4 (EML4), accounting for approximately 80% of all fusion events, resulting in the ability to directly form an ALK dimer without activation of an exogenous ligand, thereby activating ALK and its downstream RAS/ERK/STAT3/mTOR and other signaling pathways. Finally, through the promotion of cell proliferation and invasion and the inhibition of apoptosis, it promotes oncogenesis [9].

The pathogenesis of ROS1-rearranged tumors is remarkably similar to that of ALK-positive NSCLC. The prevalence is estimated to be 0.6% to over 3%, occurring primarily in non-smokers and women.

ROS1 is an “orphan” receptor tyrosine kinase with no known ligand. The gene encoding for ROS1 (c-ros oncogene 1) is located on chromosome 6 long arm at 6q22. The final protein is a member of the insulin receptor family, which is activated downstream in cells and promotes STAT3, PI3kinase, and RAS/RAF/MAPK pathways by phosphorylation of RAS [10].

The physiological functions of ROS1 are unclear, but in oncogenic terms, it seems likely that the downstream effects are similar to those exerted by the oncogenic activation of ALK.

Currently, three generations of ALK/ROS1TKI are available for clinical use.

The first oral ALK TKI approved for the treatment of ALK-positive NSCLC, crizotinib, initially showed encouraging results in the Phase 3 PROFILE 1014 study and resulted in drug use in patients with ALK-positive metastatic NSCLC [11].

However, the major issue that has emerged is secondary resistance to crizotinib, which usually occurs within a year of treatment and includes secondary ALK kinase domain mutations, *ALK* gene amplification, bypass of downstream signaling via EGFR, and drug resistance due to sub-optimal central nervous system (CNS) exposure [12].

Second-generation ALK TKIs such as alectinib [13], ceritinib [14], and brigatinib [15] have been developed to overcome therapy-induced acquired resistance. They have been shown to increase objective response and central nervous system penetrance and to prolong median progression-free survival (mPFS), becoming the gold standard in first-line treatment for ALK+ stage IV.

Although second-generation ALK inhibitors have shown remarkable efficacy in these settings, all patients virtually progress with ALK-dependent resistance, accounting for approximately 50% of cases [16].

The management of patients with advanced ROS1-rearranged NSCLC relies on first-line crizotinib, ceritinib, or entrectinib that demonstrated a mPFS ranging from 6 to 20 months. Analogously, all patients with ROS1 NSCLC eventually show tumor progression due to resistance mechanisms such as kinase domain mutations [17]. 

This clinical context prompted the urgency to design, identify, and develop more potent ALK inhibitors.

Lorlatinib (PF-06463922) is an oral small-molecule ATP-competitive inhibitor of receptor tyrosine kinases, anaplastic lymphoma kinase (ALK), and C-rosc oncogene 1 (ROS1) specifically designed to enter the central nervous system and overcome known secondary resistance mutations in the ALK tyrosine kinase domain [18].

## 3. Lorlatinib: Preclinical Development, Pharmacodynamic and Pharmacokinetics

Lorlatinib (PF-06463922) is a third-generation oral, reversible, ATP-competitive macrocyclic ALK and ROS1 TKI 18. By inhibiting the tyrosine kinase activity in a dose-dependent fashion, this drug blocks the signal transduction pathway, otherwise resulting in cell proliferation and survival [19].

The pharmacokinetic and pharmacodynamic characteristics of lorlatinib are closely related to its chemical structure.

Its structure has been optimized to confer chemical–physical properties allowing better absorption, distribution, metabolism, and excretion (ADME) and a low propensity for p-glycoprotein 1 (p GP1)-mediated efflux. Of note, the lower affinity for pGP1 and the peculiar lipophilicity of lorlatinib (expressed by LipE, a numerical index of binding effectiveness per unit lipophilicity) result in higher penetration across the blood–brain barrier (BBB) and greater efficacy on brain metastases [20].

Moreover, the macrocyclic moiety confers effectiveness in patients who have relapsed after first or second-generation TKI therapy, hence overcoming ALK resistance mutations (i.e., the most characterized “gatekeeper” variants L1196M, G1269A, and G1202R) [20].

Conformational restriction, a consequence of macrocyclization, improves the potency of the compound by minimizing the entropic cost, reduces conformational freedom, and traps the molecule in a bioactive conformation, resulting in a gain in selectivity and potency. Therefore, the macrocyclic structure allows a more effective binding between the drug and the tyrosine kinase portion of its receptor [21].

The potency and selectivity of lorlatinib were evaluated in phosphorylation and biochemical assays. In fact, phosphorylation assays performed on murine cell line NIH-3T3 showed improved potency of lorlatinib compared to crizotinib in wt-*ALK* (about a 62-fold increase). Moreover, considering *ALK* punctiform mutations (such as L1196M, G1269A, or G1202R), lorlatinib showed a 40–825-fold potency improvement compared to crizotinib [20]. Furthermore, the selectivity of lorlatinib for ALK and ROS1 kinases was evaluated in biochemical assays against a panel of 206 recombinant kinases. Of these, only 10 off-target kinases showed enzyme-based activity and selectivity margins of less than 100-fold compared to the ALK-1196M gatekeeper mutant [20].

The international regulatory agencies recommended a daily dose of 100 mg based on the following pharmacokinetic characteristics:

A single-dose plasma peak concentration is achieved with a median Tmax of 1.2 h. The steady-state plasma half-life of lorlatinib is 14.83 h. In vitro, 66% of lorlatinib is bound to plasma proteins. Regarding metabolism, lorlatinib undergoes oxidation and subsequent glucuronidation conjugation, and it is mainly metabolized by CYP3A4 and UGT1A4 enzymes, with minor involvement from CYP2C8, CYP2C19, CYP3A5, and UGT1A3. The concomitant use of lorlatinib with some potent inducers (such as rifampicin) or inhibitors (such as Itraconazole) of the enzymes involved in its metabolism can modify its blood concentration, efficacy, and toxicity.

Therefore, concomitant use of lorlatinib with potent CYP3A4/5 inducers is contraindicated. Conversely, in the case of concomitant use of lorlatinib with potent CYP3A4 inhibitors, a dose reduction of lorlatinib should be considered.

Lorlatinib is also a moderate inducer of CYP3A and a weak inducer of UGT1A; therefore, its use with other CYP3A4/5 substrates with narrow therapeutic indices should be avoided, while no dose adjustment is necessary in the case of concomitant use of lorlatinib with drugs that are substrates of UGT1A.

Lorlatinib is extensively metabolized in the liver, even if there is no information available regarding its use in patients with moderate or severe hepatic impairment [19].

By contrast, a reduced dose of lorlatinib (from 100 mg daily to 75 mg daily) is recommended in patients with severe renal impairment [22].

In fact, with regard to the excretion, following a single dose of radiolabeled lorlatinib, 41% of the radioactivity was excreted in feces (less than 1% was unchanged), and 48% was excreted in urine (approximately 9% was unchanged) [23].

## 4. Place in Therapy of Lorlatinib for ALK Rearranged NSCLC

How lorlatinib fits into the therapeutic algorithm for ALK-rearranged NSCLC remains unclear. The Food and Drug Administration (FDA) approved lorlatinib for first-line treatment in 2021 [24]. In early 2022, it was approved as the first line in the US, Europe, Japan, and China.

Lorlatinib was first studied in an international, multicentre, single-arm first-in-man phase 1 trial that showed its systemic and intracranial activity in patients with ALK- or ROS1-rearranged NSCLC, previously treated with ALK/ROS1 TKIs. The overall response rate (ORR) was 46% among all ALK-positive patients and 42% among those who previously received ≥2 TKIs. The maximum dose tolerated was not identified, and the dose approved for the phase 2 trial was set at 100 mg once daily [18].

In the phase 2 trial, patients were enrolled in six different cohorts based on ALK and ROS1 status and prior line of therapy; the activity of lorlatinib was confirmed both in patients previously treated with TKIs, with an ORR of 47% and an objective intracranial response of 63% (in patients with measurable baseline CNS lesions); even in naïve patients, an objective response was obtained in 90–95% of patients. Three of these patients had baseline CNS lesions, and two of them had an objective intracranial response [25].

The CROWN trial was a phase 3 study that compared lorlatinib with crizotinib in patients with metastatic ALK+ NSCLC. After a median follow-up time of 18.3 months, the median progression-free survival (PFS) was not reached (95% CI, NR-NR) with lorlatinib and 9.3 months (95% CI 7.6–11.1) with crizotinib (hazard ratio [HR] 0.28 [95% CI 0.19–0.41 *p* < 0.001]. In a recent update, the 3-year PFS rate was 64% (95% CI, 55–71) in the lorlatinib group and 19% (95% CI, 12–27) in the crizotinib group [26].

Lorlatinib was designed to cross the BBB, with significant clinical implications for patients with brain disease. Solomon et al. demonstrated that one-year lorlatinib treatment was associated with less brain progression than crizotinib in patients with (7% vs. 72%) and without (1% vs. 18%) brain metastasis at the baseline [27]. In addition, lorlatinib is an effective option after progression to a first- or second-generation ALK-TKI. Shaw et al. showed that lorlatinib, unlike previous-generation ALK-TKIs, significantly inhibited ALK phosphorylation in secondary *ALK* mutations, including the G1202R [16]. However, the lack of studies directly comparing lorlatinib to second-generation ALK-TKI limits its definitive indication as the preferred first-line treatment. A meta-analysis including previously untreated patients with ALK+ NSCLC was conducted with the primary aim of providing a comparison between lorlatinib and alectinib in terms of efficacy and safety. The results showed no significant differences in PFS and OS; nevertheless, lorlatinib obtained a surface under the cumulative ranking curve (SUCRA) of PFS superior to alectinib (97.4 vs. 79.2). The ethnic subgroup analysis, instead, showed that in non-Asians patients, PFS was significantly more favorable with lorlatinib than alectinib, which is more effective in Asians [28].

Despite the first-line indication, the absence of mature survival outcomes, the safety profile, and the lack of sequential treatments may weaken the upfront use of lorlatinib. Remarkably, the initial TKI choice will influence the tumor’s molecular evolution and subsequent therapeutic options.

In the case of oligoprogression, locoregional treatment should be considered. An Italian multicentric analysis that investigated the activity of radiotherapy in patients with ALK-/EGFR-mutated metastatic NSCLC confirmed that stereotactic RT (HR 0.355, CI 95% 0.212–0.595; *p* < 0.001) and median duration of TKI superior to 14 months (HR 0.17, 95% CI 0.10–0.30; *p* < 0.001) were independent factors related to better OS [29].

When the progression is not amenable to local treatment, switching to further treatment is mandatory. ESMO guidelines recommend ChT with a platinum–pemetrexed-based combination, but other alternatives are possible [30].

In the IMpower150 trial, the benefit of the atezolizumab + bevacizumab + carboplatin + paclitaxel combination observed in patients with *EGFR* or *ALK* genetic alterations was notable, with a mPFS of 9.7 months in the immunotherapy arm (95% CI, 6.9–15.2) versus 6.1 months (95% CI, 5.7–8.4) in patients treated without atezolizumab [31]. Nevertheless, ALK-positive patients were underrepresented compared to EGFR-positive patients, and no formal analyses have been conducted in this specific subgroup, limiting the evidence of effectiveness.

A retrospective analysis in the ongoing phase 2 trial has shown the efficacy of lorlatinib beyond progression in patients pretreated with crizotinib alone or with at least one II-generation TKI. The authors showed that continuing lorlatinib beyond progression instead of ALK-TKIs, without other therapies or radiotherapy, determined a benefit in terms of OS that was greater in the group previously treated with crizotinib [32,33].

The study of IV-generation ALK inhibitors is under investigation to overcome resistance mechanisms to lorlatinib treatment. TPX-0131 and NVL-665 are two new drugs that are the topic of phase I studies, but more data are needed [34,35,36].

The therapeutic algorithm for ALK-rearranged NSCLC, according to literature evidence and international guidelines, is illustrated in Figure 1. Table 1 resumes the main clinical studies investigating the use of Lorlatinib in different clinical settings.

## 5. Place in Therapy of Lorlatinib for ROS1 Rearranged NSCLC

Available targeted therapeutic options for patients with NSCLC harboring ROS1 rearrangements are currently limited. Compounds indicated in the first-line setting are crizotinib and entrectinib, preferred over crizotinib in patients with brain metastases, and repotrectinib, which is neither approved by the EMA nor the FDA [30].

Crizotinib was the first TKI approved for ROS1+ NSCLC, following the positive results of the PROFILE 1001 clinical trial. The study involved 50 patients with ROS1–NSCLC, of whom the majority (86%) had received at least one previous line of standard therapy for advanced NSCLC and demonstrated high efficacy with an ORR of 72% [54].

In the 2019 study update, survival data were mature, with a median OS of 51.4 months and a 79% 12-month survival probability. Responses were durable, with a median duration of response (DoR) of 24.7 months and a median PFS of 19.3 months [55]. 

Unfortunately, all crizotinib-treated ROS1-positive patients will virtually incur disease progression due to on-target mutations (the most common is Gly2032Arg, which sterically impedes the compound binding [56,57]), off-target mutations, and CNS progression [58]. 

Based on guidelines from ESMO and NCCN, lorlatinib can be considered as a treatment option for patients with metastatic ROS1–NSCLC who have previously received crizotinib, ceritinib, or entrectinib. However, neither the FDA nor the EMA have yet approved lorlatinib for this use [30,59].

The clinical efficacy of lorlatinib against ROS1–NSCLC was investigated in the multicenter NCT01970865 study, which included 69 ROS1-positive advanced NSCLC patients with no restrictions regarding CNS metastases [25,37]. A total of 40 patients received prior crizotinib, 8 previously received one or more non-crizotinib TKIs, and 21 were TKI-naïve. Lorlatinb led to a rapid and long-lasting response in both crizotinib-pre-exposed and crizotinib-naïve patients. The median DoR was 25.3 months in TKI-naive and 13.8 months in crizotinib pretreated, and the ORR was, respectively, 61.5% and 26.5%. Among the 25 patients with brain metastases, the ORR was 52.6% in crizotinib-pretreated patients and 66.7% in crizotinib-naïve patients. Notably, lorlatinib appears to be more effective in TKI-naive patients [37]. Further analysis showed that lorlatinib was ineffective in patients with the G2032R mutation, which could justify this difference [37]. G2032R is one of the most common ROS1 on-target mutations after treatment with crizotinib [56,60]. Although in vitro studies reported promising results on the ability of lorlatinib to overcome this resistance mutation [61,62], preclinical activity does not translate into clinical efficacy. The Italian PRFOST study and the NCT01970865 study demonstrated a poor capacity of lorlatinib in controlling disease in ROS1 G2032 mutated NSCLC with no disease response [37,63]. On the other hand, the acquisition of a ROS1 G2032R mutation was also documented during treatment with lorlatinib [57,64,65,66].

The GLASS study is a retrospective trial involving 123 patients (including 17 patients with ROS1–NSCLC and 106 patients with ALK rearrangement-NSCLC) that has demonstrated the efficacy of lorlatinib in controlling both intra- and extracranial disease with an ORR of 67% in patients receiving lorlatinib as second-line (pretreated with other TKIs or with chemotherapy) and an ORR of 100% in patients receiving lorlatinib as third-line [38].

The LORLATU study, derived from the French expanded access program, was the largest study that evaluated the treatment sequence in ROS1–NSCLC after the failure of at least one ROS1 TKI. Lorlatinib’s safety and efficacy were specifically investigated as secondary objectives. The trial enrolled 80 patients, of whom 51 had intracranial metastases at lorlatinib initiation. Remarkably, the median PFS was 7.1 months, with an intracranial response of 72% and an ORR of 45%. A noteworthy response has been documented in patients pretreated with at least one ROS1-TKI. In particular, the study showed a median PFS of 8.5 months in crizotinib-pre-exposed patients, an ORR of 35%, and an intracranial response of 50%. This evidence allowed the authors to state that lorlatinib may be a valuable therapy in patients resistant to at least one ROS1-TKI [39].

A phase-2 study evaluated the intracranial response in patients with only brain progression to crizotinib (both parenchymal and leptomeningeal and ± previous radiotherapy treatment). The study showed notable efficacy of lorlatinib in controlling brain disease, with an intracranial ORR of 87% and a complete intracranial response in 60% of patients. The median intracranial PFS was 38.8 months, and the extracranial PFS was 41.1 months [40].

Although the data related to ROS1-driven NSCLC and lorlatinib are limited, resistances to lorlatinib have been identified since early trials [25]. Resistance mechanisms to lorlatinib are not yet fully understood. On-target mutations have been documented as causes of drug resistance. Lorlatinib use is not appropriate in cases of ROS1 G2032R and ROS1 L2086F mutations, which confer resistance to the drug. In G2032R-mutated patients in progression to first-line therapy, clinicians should evaluate new promising drugs like repotrectinib and taletrectinib, while the L2086F mutation predicts a response to cabozantinib [66,67]. 

In conclusion, considering the literature so far, the therapeutic validity of lorlatinib in patients affected by ROS1 NSCLC can be affirmed. Nevertheless, it is not sufficient to define the therapeutic setting of this drug in a scenario lacking comparative studies. 

A hypothetical mutation-driven therapeutic algorithm for ROS1-rearranged NSCLC is illustrated in Figure 2. 

## 6. Resistance Mechanisms to ALK Inhibitors

ALK-TKIs have been shown to be a strong therapeutic option for patients with ALK- or ROS1-rearranged NSCLC; nonetheless, the emergence of resistance mechanisms still represents the main obstacle to overcome. The attempt to overcome heterogeneous resistance mechanisms has led to the development of different generations of ALK TKIs, from crizotinib to lorlatinib. However, this issue will not be progressively extinguished since the selective pressure of personalized therapy favors the onset of novel resistance pathways. It is possible to identify several mechanisms of ALK-TKI resistance. 

Resistance mechanisms to ALK-TKIs can be classified according to the time of onset (de novo vs. acquired) or according to the involvement of the ALK receptor (on-target vs. off-target).

ALK-dependent (or “on-target”) mechanisms are the most common cause of acquired secondary resistance, accounting for 80% of ALK TKI treatment failure, 30% during crizotinib therapy, and 50% during next-generation inhibitors. The most common genetic alterations are ALK secondary mutations in the tyrosine kinase domain, blocking the ligand-receptor interaction, but also amplifications or loss of the ALK fusion. The relative frequency of these epigenetic variants has proven to increase passing through different generation drugs significantly: resistance mutations have been detected only in about 20% of patients progressed during crizotinib treatment, while 54%, 53%, and 71% of patients developed resistance after ceritinib, alectinib, and brigatinib therapy, respectively [16,69,70,71,72]. Resistance to ALK TKIs mostly depends on secondary mutations occurring at residues 1151, 1152, 1156, 1174, 1202, 1203, 1206, and 1171. Among them, G1202R is the most common secondary mutation, known to cause resistance to crizotinib, ceritinib, and alectinib but maintain sensitivity to brigatinib and lorlatinib [16,69,73,74,75]. Based on the receptor area where the mutations occur, we can identify gatekeeper mutations (e.g., L1196M), solvent-front mutations (e.g., G1202R, D1203N, S1206), other second-site mutations (e.g., G1269A, 1151Tins, L1152R, C1156Y, F1174C), and compound mutations (e.g., G1202R-containing compound mutations, C1156Y + L1198F, I1171S + G1269A) [16,70,76,77,78,79,80,81].

ALK-independent or “off-target” mechanisms arise in cases of genetic alterations of other genes different from ALK and in the presence of a perturbation of the homeostatic signaling transduction routes. They include a broad range of escape strategies that go from the activation of downstream or parallel signaling routes, like PI3K–AKT, MAPK, KIT, IGF-1R, EGFR, NRAS, FGFR, KRAS, MEK, and MET, to the histological transformation, passing through the epithelial to mesenchymal transition and the drug-efflux [71,72,78,80,82,83,84,85,86,87,88,89,90,91,92,93,94,95].

Virtually all patients will develop resistance to lorlatinib at some point due to the emergence of resistance mechanisms. The acquisition of mutations in the tyrosine kinase domain of the same ALK allele, also defined as cis compound mutations, determines around 35% of disease progression cases during lorlatinib treatment [77,96,97,98,99,100,101]. Several compound mutations have been observed in lorlatinib resistance clones [101]. The most common ones are C1156Y + L1198F [100], I1171N + L1198F, L1171N + D1203N, L1196M + G1202R, G1202R + G1269A, D1203N + E1210K, G1202R + L1204V + G1269A, D1203N + E1210K + G1269A [77,96,102], F1174L + G1202R, T1151M + G1202R [98], I1171N + D1203N (see Table 2) [96]. Since the G1202R variant is often found in patients progressing to II-generation ALK inhibitors, we can expect a prevalence of compound G1202R mutations [16]. Some of the known compound mutations have also been shown to determine re-sensitization to ALK inhibitors of prior generations [77,99,100,101]. In fact, any compound mutation comprising the L1198F, in particular ALK-I1171N + L1198F, C1156Y + L1198F, and G1202R + L1198F, is predictive of crizotinib sensitivity, de novo or restored [100]. Similarly, alectinib has proven to be effective against the ALK-I1171N + L1256F lorlatinib double resistance mutation. Ceritinib can still control disease after ALK-I1171N + L1196M and I1171N + G1269A compound mutations, and finally, brigatinib is active in cases of I1171N + L1198F, +L1196M, +L1256F, and + G1269A mutants [97].

Different strategies and known molecules have been tested to overcome lorlatinib resistance determined by compound mutations. Some authors have focused their attention on resistance induced by I1171N compound mutations (I1171N + L1198F, I1171N + L1256F, or I1171N + L1196M) [121]. They found that gilterinib, a multi-kinase inhibitor approved for treating AML in Europe, the USA, and Japan, was effective in this disease subtype in pre-clinical models [122]. Nevertheless, gilterinib also exerts inhibitory control on different downstream signaling pathways, so the inappropriate activation of these routes as bypass mechanisms can induce drug resistance. This resistance could be partially controlled by the combination of gilterinib and RTK-specific target therapies [121]. 

In some cases, no secondary ALK mutations are detectable at disease progression under treatment with lorlatinib. In these cases, we can consider other off-target mechanisms of resistance (see Figure 3). MET and EGFR pathway hyperactivation is the most frequent ALK-independent resistance mechanism described in the literature. 

The constitutional activation of MET in these cases can depend on mutations, copy number variations, ligand overexpression, or fusions. Both in ALK-rearranged and EGFR-mutated diseases, it is possible to establish a correlation between the probability of MET amplification and the TKI’s power and specificity, with this genetic alteration being more common after III-generation drugs. In the study by Dagogo and colleagues, MET amplifications were identified in 12% of specimens collected from patients progressing to II-generation TKIs and in 22% of biopsies made to recharacterize disease at lorlatinib progression. Considering its multi-kinase inhibitor nature, no cases of MET amplification were present in samples of patients pretreated with crizotinib. Then, exposure to crizotinib could be seen as a protective factor against MET dysregulation [123,124,125]. In the subset of patients with MET-driven resistance, some researchers are evaluating the potential efficacy of the dual combination of ALK/MET TKI, associating lorlatinib with capmatinib, crizotinib, or savolitinib. Since capmatinib is more capable of passing the BBB, the combination therapy of capmatinib and lorlatinib could be a suitable option for MET-induced progression in ALK + NSCLC [11,125,126]. EGFR expression has a negative prognostic role in ALK-rearranged NSCLC [127,128]. The interplay between ALK and EGFR derives from the lorlatinib-induced c-Jun-mediated increased expression of HB-EGF, which causes the phosphorylation of EGFR, AKT, and ERK, sustaining the secondary resistance to the drug through this pathway. From here derives the rationale for the use of EGFR TKIs, like erlotinib, in association with lorlatinib [129].

In some patients with a negative mutational pattern, researchers have identified p53 mutations, the downregulation of the NF2 gene, and the hyperexpression of mir-100-5p, leading to chromosomal instability, mTOR proliferative pathway stimulation, and epigenetic regulation, respectively. Some therapeutic options have been proposed in these cases: the use of the p53 activator APR-246, combined therapy with lorlatinib plus mTOR or multi-kinase inhibitors (everolimus and vistusertib), and synthetic LNA molecules against miRNA overregulation [99,130,131].

A role is also recognized for the epithelial-mesenchymal transition (EMT) [99,132]. The association of SRC inhibitors (saracatinib or dasatinib) with lorlatinib has been shown to have efficacy on mesenchymal cells, gradually increasing their sensitivity to the cytotoxic action of lorlatinib [99]. 

To complete this overview, we must mention the tumoral histological change from adenocarcinoma to squamous cell lung cancer (SqLC). ALK-rearranged SCLC has a notoriously lower response to TKIs than ALK-rearranged adenocarcinoma [133]. For this reason, the histological transition can be interpreted as a negative prognostic factor [134]. In EGFR-mutated NSCLC, there was a relationship between the histological change and the female sex, the persistence of the EGFR mutation, and the status of former smokers [135]. Also, several cases of adenocarcinoma to small cell lung cancer (SCLC) have been documented in the literature. This cellular phenotype change seems to be correlated with female sex, an intermediate age, Asian ethnicity, and nonsmoking status. The histological transition can be recognized through the acquired expression of synaptophysin, chromogranin, or CD56, the loss of RB1, and the p53 mutation. The incidence of the NSCLC-SCLC transformation progressively increases with drug generation. Even using standard chemotherapy schedules, alone or in combination with TKIs, the prognosis of these patients remains substantially unfavorable [136].

As for ALK, mechanisms of resistance to ROS1-TKIs can be categorized into ROS1-dependent and ROS1-independent. Mutations in the tyrosine kinase domain of the ROS1 gene are responsible for 43% of ROS1-rearranged NSCLC patients acquiring resistance to lorlatinib [66,137]. The most common are the G2032R [56,66,138,139,140] and the L2086F [66]. To overcome the resistance to the standard personalized therapies of I line (crizotinib, entrectinib) and II line (lorlatinib and ceritinib) in this setting, several treatments have been proposed, including cabozantinib, repotrectinib, taletrectinib, and ensartinib [20,137,140,141]. Less is understood about the ROS1 bypass mechanisms of resistance. Alterations leading to the functional hyperactivation of other RTKs (MET, EGFR, and KIT [142,143,144]) or of downstream pathways (NF1, BRAF, MEK, and MAPK) have been reported [145,146]. Recent studies have also identified mutations in TSC1/2, SMAD4, CTNNB1, APC, NOTCH1, PTCH1, CDKN2A, CCNE1, ATM, and BRCA1/2 [66]. In about 20% of all cases of lorlatinib resistance, it has been identified as an amplification of the MYC proto-oncogene that, in pre-clinical studies, has been demonstrated to induce resistance to all TKIs employed in ROS1-mutated disease [147]. Since MYC has a stimulatory function over different genes involved in proliferation, like cell cycle kinases like CDK4 and 6, this could lead to the testing of CDK4-6 inhibitors as a therapeutic alternative [147]. 

**Figure 3 diagnostics-14-00048-f003:**
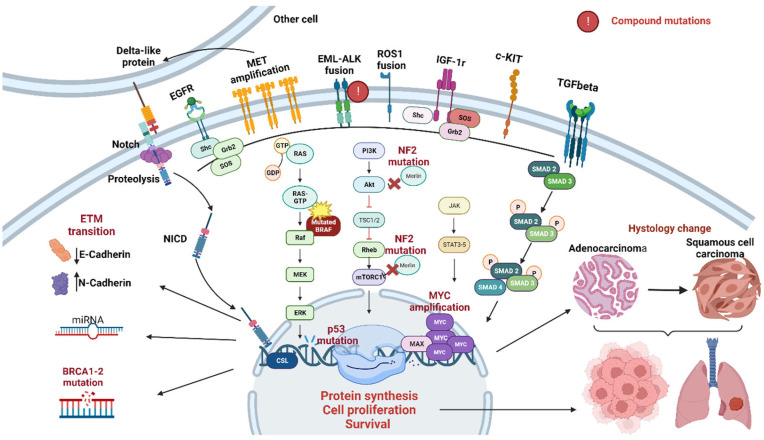
Off-target resistance mechanisms to III generation lorlatinib [66,71,72,78,80,82,83,84,85,86,87,88,89,90,91,92,93,94,95,99,123,124,125,127,128,130,131,132,133,145,146,147].

The number of potentially targetable resistance strategies underlines the importance of repeating biopsy at progression to TKIs [100]. In detecting resistance mutations, plasma genotyping and circulating DNA (cDNA) are gaining increasing importance, being more informative about genetic modifications than solid biopsy, with the advantage of a non-invasive procedure [42,148,149,150,151]. In fact, solid biopsy has an intrinsic risk and can give only a small amount of material from a single tumoral location, while circulating DNA is representative of the molecular bioprofile of malignant cells from all the metastatic disease sites [102,152,153].

## 7. Safety Profile and Clinical Management

Lorlatinib treatment should be continued until there is unacceptable toxicity or evidence of disease progression. The recommended daily dosage is 100 mg [18], with a potential stepwise posology reduction first to 75 mg daily and, in cases of persistent suboptimal tolerability, to the minimum efficacious dosage of 50 mg. The dose reduction does not correspond to a mirrored reduction in therapeutic power, with a clinical benefit demonstrated for the lower dosage [154]. Notably, the post hoc analysis of the CROWN study [26] demonstrated that the 12-month PFS of patients with and without dose reductions were comparable [27,155,156].

Despite its clinical efficacy, lorlatinib is characterized by a suboptimal safety profile, which has earned it a special medical warning. The safety concern regarding lorlatinib treatment depends not only on the possible seriousness of AEs but also on their heterogeneous entity. The safety analysis of the phase III CROWN trial evidenced a different toxicity profile between lorlatinib and crizotinib. Lorlatinib was associated with a higher incidence of hypercholesterolemia (70% vs. 4%), hypertriglyceridemia (64% vs. 6%), edema (55% vs. 39%), increased weight (38% vs. 123%), peripheral neuropathy (34% vs. 15%), and CNS AEs (21 vs. 6%). Analogously, anemia (19% vs. 8%), hypertension (18% vs. 2%), and hyperlipidemia (11% vs. 0%) were more common in the lorlatinib cohort. On the contrary, crizotinib showed a stronger association with gastrointestinal events like diarrhea (52% vs. 21%), nausea (52% vs. 15%), vomiting (39% vs. 13%), and constipation (30% vs. 17%), but also with increased alanine and aspartate aminotransferase (61% vs. 31% with a prevalence of ALT raises), fatigue (32% vs. 19%), anorexia (25% vs. 3%) in part secondary to dysgeusia (16% vs. 5%), ocular symptoms (39% vs. 18%), and bradycardia (12% vs. 1%). Apart from the diversity of AEs, the crizotinib toxicity profile was more favorable regarding the probability of grade 3–4 events, which occurred in 56% of crizotinib and 72% of lorlatinib-treated patients, respectively. They consisted especially of alterations of triglycerides (20%) and cholesterol (16%), hypertension (10%), and weight increases (17%) [41].

Focusing on the toxicity profile of lorlatinib, hyperlipidemia (81% hypercholesterolemia and 60% hypertriglyceridemia), increased weight (18%), peripheral neuropathy (30%), edema (43%), and cognitive impairment were the most common side effects that emerged in the CROWN trial and were confirmed in everyday clinical practice [25,39,41,43]. In line with the CROWN trial and also in other studies, diarrhea (11%), nausea and emesis (5%), ALT (9%), AST (11%), and bilirubin increased, heart rate alterations (bradycardia), and anorexia, typical AEs of other ALK inhibitors like crizotinib, are less frequently referred to during lorlatinib therapy [25,41]. In the phase II study by Solomon et al. on ALK-rearranged NSCLC, 30% of patients needed a dose interruption, and for 22% of them, a dose reduction was necessary [25]. Oedema was the AE leading to the greater number of dose interruptions (6%) and dose reductions (7%) in the study cohort. Nevertheless, the rate of patients who required permanent discontinuation was low (3%) and mostly related to the onset of neurocognitive impairment. Hypercholesterolemia and hypertriglyceridemia were the most frequently seen G3–4 AEs (32% of the population) [25]. Going deeply into the most significant collateral event, in the safety phase II study, hypercholesterolemia was detected in 81% of patients, with 66% of G1–G2 events, 14% of G3 increases, and only 4 cases of G4. Hypertriglyceridemia affected 61% of the population, with a predominance of G1-G2 events (45%) in comparison to G3 (13%) and G4 (3%) ones [25]. In the study by Shawn et al. of lorlatinib in ROS1-rearranged naïve and pretreated patients, 96% of subjects had at least one adverse event, with a prevalence of hypertriglyceridemia and hypercholesterolemia. 36% of patients underwent treatment discontinuation, and a minor percentage, 25%, assumed a reduced drug dosage after the onset of lipidic alteration, edema, and peripheral neuropathy. Also, in this case, no drug-related deaths were reported, and only one patient was obliged to interrupt the assumption because of severe AE [37]. 

In consideration of the great therapeutic potential and the good overall survival guaranteed by lorlatinib in the NSCLC advanced adenocarcinoma scenario, the identification of proper management of therapy-related AEs is gaining growing interest [157,158]. 

The mechanisms underlying the altered lipid profile in patients treated with lorlatinib are still uncertain. It has been supposed that this drug can alter the mitochondrial fatty acid metabolism in hepatic cells, leading to high lipidic intracellular concentrations and steatohepatitis [159,160,161]. In addition, there is literature evidence about lorlatinib’s potential to cause renal damage in the form of nephrotic syndrome. Nephrotic syndrome is associated with hyperlipidemia, so lipidic imbalance could also result from concomitant renal impairment [162,163,164,165].

After three weeks of continuative treatment, hyperlipidemia and hypercholesterolemia reach their maximum incidence. For this reason, lipidic status should be tested at baseline, prior to starting therapy, and then routinely monitored during active treatment. The approach to high lipidic levels (a common warning threshold above 300 mg/dL for triglycerides and cholesterol) forecasts introducing hypolipidemic drugs like statins and fenofibrate [166]. Since cytochrome CYP34A family enzymes metabolize lorlatinib, choosing between pravastatin and rosuvastatin should be preferable for the lower pharmacokinetic interactions [167]. Fibrates like fenofibrate at the daily dosage of 200 mg, fish oils, and ezetimibe (10 mg daily) can be added [168,169]. Each patient’s cardiovascular profile should be considered, considering primary prevention with pravastatin 20 mg or rosuvastatin 5 mg [170]. Secondary prevention requires higher dosages, such as pravastatin 40 mg and rosuvastatin 20–40 mg. Even after grade four hyperlipidemia, there is the possibility to restart treatment at the full dosage if toxicity regresses to G2 or a lower degree. A dose de-escalation must be considered only in cases of recurrence of high-grade events, conventionally defined by cholesterol > 500 mg/dL and triglycerides >1000 mg/dL [171]. 

Silibilin is a flavonolignan that modulates lipidic homeostasis, avoiding cholesterol and triglyceride accumulation. In some case reports, silibilin has shown a potential coadjutant role in the control of brain metastatic disease, so the association of silibilin with lorlatinib could improve the management of patients with CNS disease, attenuating one of the lipidic profile impairments [172,173,174,175]. 

Weight gain associated with increased appetite and food intake deserves particular attention, requiring nutritional support if needed. The highest median weight increase reported was approximately 11% from the baseline. However, this collateral event has been recognized only in 20% of patients, with no relevant effect on the treatment [169].

An impairment of the central or peripheral nervous system (PNS) is a burning issue in managing lorlatinib-related AEs. A total of 39% of patients who received lorlatinib treatment experienced neurological and cognitive symptoms [25]. Similarly, the post hoc analysis of the CROWN study confirmed that 35% of patients complained of CNS AEs [27]. A plethora of symptoms are connected with lorlatinib-related neurotoxicity: mood instability (16%), sleep disorders, altered mental status with personality abnormalities and memory dysfunction (21%), speech difficulties (7%), and psychotic effects (3%) (see Figure 4) [25,27,166]. Similar incidences were reported by Solomon et al. [25]. Mood instability, arising from feelings such as anxiety, depression, irritability, and euphoria, affected 22% of patients. In 2.7% and 2.4% of cases, dose withholding and reduction were necessary, respectively. The majority of medical issues that required intervention were resolved within a timeframe of approximately 14 days. A recurrence was seen in 33.3% of patients who restarted treatment at a low dosage, while paradoxically, no rebound was noticed in patients who maintained the same posology. These symptoms are usually resolved with drug discontinuation. The incidence was comparable across age subgroups, but a major incidence has been found in the non-Asian population [25,169]. Cognitive impairment has been diagnosed in 23.1% of the study population, requiring interruption of lorlatinib in 3.7% of cases and dose reduction in 2.7%. Cognitive impairment has been described as a memory deficit or confusion associated with visual and auditory hallucinations. This neurological effect declined in most patients after dose modification for a median of 10 days. Other CNS disorders frequently reported include aphasia, slow speech, and difficulty composing full-sense phrases. Speech alterations were detected in more than 9% of patients. Only three patients underwent dose interruption and/or reduction, and the resolution of referred symptoms required a longer time interval than other CNS toxicities (38 days) [25,169]. Generally, these secondary effects are most common in the first two months of treatment. In the study by Solomon et al., 91% of patients had CSN AEs of mild–moderate entity (G1/G2) [25]. In the CROWN study, 62% of these events were resolved or attenuated without medical intervention, and only 23% needed dosage reduction [27]. In cases of grade 1 CNS toxicity, clinicians should continue the treatment or temporarily stop it and then reintroduce lorlatinib at the same or a 25% reduced dosage. Lorlatinib can be temporarily interrupted in cases of grade 2 to 3 AEs and then re-started at a low dosage after toxicity resolution or downstaging to grade 1. Continuing therapy for grade 4 adverse events is harmful and, therefore, not recommended [155,169].

An inferior incidence of neurological AEs in the Asian population has been supported by different clinical studies: in the trial by Lu et al., less than 10% of the study population experienced such cognitive manifestations, and a similar percentage was reported by Ross and colleagues [43,176,177]. 

Notably, the identification of risk factors for CNS toxicity may help the decision-making process of therapeutic strategy. The CROWN study evidenced that encephalic radiotherapy, brain metastasis, psychiatric disease, and the use of psychoactive drugs have been significantly associated with CNS toxicity. In particular, 42% of patients with brain secondary lesions versus 32% of patients without brain metastases experienced cognitive symptoms, while more than half of patients with prior RT had CNS AEs (56 vs. 34 for crizotinib) [27]. Clinicians need to carefully consider prescribing lorlatinib to patients with these risk factors. It is recommended to consult with other specialists, such as psychiatrists, and work as a multidisciplinary team. In some cases, it may be preferable to use II-generation ALK inhibitors [27,169]. 

It is important to educate family members on identifying early signs of CNS compromise and seeking immediate medical attention. 

CNS toxicity can coexist with PNS involvement. Peripheral neuropathy has a later onset than other AEs described and can be seen in almost 50% of patients. Despite the prevalence of mild manifestations, it is the second most important reason for lorlatinib interruption (4.1%) or dose modification (4.1%) after edema. This subset of patients can benefit from supplementation with group B vitamins and the use of drugs for neuropathic pain, like pregabalin or gabapentin [169].

Like other ALK-TKIs, lorlatinib can induce edema in more than 40% of treated patients, with a prevalence of peripheral edema that can be contrasted through behavioral measures (elastic compressive socks, regular physical activity). The use of diuretics is reserved for G2–3 edema, with a predilection for furosemide and spironolactone in cases of furosemide resistance. Even if it is generally mild or moderate in extension, it has a negative impact on lorlatinib therapy, making unavoidable dose reductions and therapeutic pauses [169].

Finally, it is important to cite anecdotal cases of ECG modification in the course of treatment, such as first-degree AV block and QTc interval prolongation. For this reason, a baseline ECG is recommended [169].

Notably, Dagogo et al. documented two grade 5 respiratory adverse events related to lorlatinib. In one of the two patients, chest imaging revealed a thoracic disease progression with concomitant pneumonitis. In the other case, the patient developed acute respiratory failure without evidence of other triggers like cardiovascular disease or pulmonary embolism. Both events occurred after a significant duration of treatment, respectively, 19 and 24 months [178]. This observation must encourage us not to underestimate respiratory symptoms like dyspnoea or cough, especially if they are progressively worsening. 

Interestingly, in the LORLATU study, the authors have seen 13% of severe adverse events leading to permanent lorlatinib discontinuation. This incidence was higher than in the other studies mentioned above. Among these AEs, there were two cases of renal failure, one of grade 3 e and the other of grade 4 52. This is challenging since renal damage is not usually included in the common collateral events associated with lorlatinib, while it has been reported during alectinib therapy [179,180,181]. Some case reports described proteinuria induced by lorlatinib, even if it is unclear if renal toxicity is the secondary effect of lorlatinib or previous TKI inhibitors [41,182]. 

Similarly, the LOREALAUS Australian real-life data study shows a strikingly high incidence of thromboembolic events diagnosed in 23 patients [45]. It is not clear if this can be considered a real side effect of ALK target therapy or if it depends more on the peculiar biomolecular characteristics of ALK-rearranged neoplasia. A 3–5-fold increase in TVE in ALK and ROS1 rearranged NSCLC has been documented in different trials [183,184,185,186,187]. 

Several indirect safety comparisons among different generations of TKIs are available in the literature. Globally, grade 3 adverse events were more frequent with lorlatinib than with alectinib and crizotinib. Brigatinib shows results similar to the III-generation TKI [28] and has a less favorable adverse event profile in comparison to crizotinib, with a frequency of 73% vs. 61% of grade 3 complications [188]. Alectinib was the only ALK inhibitor to show a safety benefit over crizotinib, with a frequency of grade 3 events of 45% vs. 51% [189]. Lorlatinib seems to be safer than ceritinib [190,191]. Compared to pemetrexed-based chemotherapy, lorlatinib shows an unfavorable toxicity profile since pemetrexed was responsible for no dose interruption or de-escalation, and a minor rate of grade 3 events occurred (14% vs. 38%). In the review by Tao and colleagues focused on ALK-TKIs toxicities, every patient included had at least one AE, and the probability of grade 3 AEs was slowly different in comparison to other data, with a probability of 72.8% for brigatinib, 72.4% for lorlatinib, 71.3% for ceritinib, 44.6% for crizotinib, and 37.4% for alectinib [192]. An indirect comparison among the most common systemic AEs during treatment with different generations of ALK TKI based on phase II trial tolerability analysis is represented in Figure 5.

In conclusion, lorlatinib has a manageable safety profile since most adverse events are mild, requiring only temporary dose interruption, dose reduction, or support therapy. A definitive treatment suspension is rare [25]. The use of orlatinib in clinical practice is also supported by a referred improvement in quality of life regarding some invalidating symptoms like insomnia, fatigue, anorexia, dyspnoea, cough, and pain. An advantage was seen in social roles, physical wealth, and the psychological sphere and was maintained over the course of treatment. Patients with brain metastases at diagnosis benefited the most from lorlatinib treatment. The worsening of baseline health conditions seen in some patients was generally due to the classical and more disturbing AEs of the drug, like cognitive impairment and peripheral neuropathy [25,197].

## 8. Conclusions

Lorlatinib is a third-generation macrocyclic ALK inhibitor able to overcome resistance to I and II-generation ALK TKIs with good clinical efficacy on systemic disease and intracranial metastases thanks to high BBB penetration [26,30].

In the end, what is lorlatinib’s future? In ALK-positive disease, lorlatinib seems to be superior to alectinib in terms of PFS, OS, and ORR in indirect comparative studies, but a III-phase trial of alectinib versus lorlatinib in patients with ALK-rearranged NSCLC in the I line is not available yet. And it is reasonable since introducing lorlatinib as first-line treatment for metastatic patients would increase their probability of having an optimal disease response but, on the other hand, would limit the therapeutic strategies at progression. 

In ROS1-rearranged disease, lorlatinib has shown similar efficacy to crizotinib and entrectinib as first-line therapies. Nevertheless, it is usually considered a second-line option based on its capability to overcome some of the resistance mechanisms to crizotinib and on the narrow therapeutic armamentarium in this setting of patients [37]. Table 3 resumes the ongoing trials investigating the efficacy of Lorlatinib among ALK or ROS1-rearranged patients.

## Figures and Tables

**Figure 1 diagnostics-14-00048-f001:**
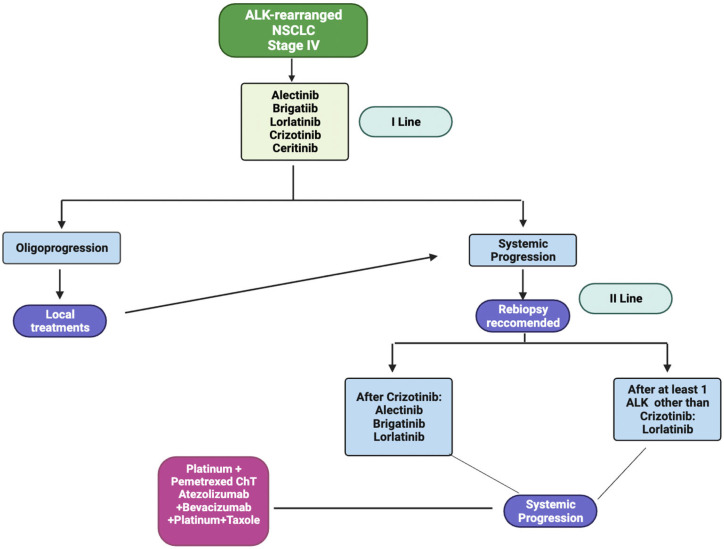
Therapeutic algorithm for locally advanced or metastatic ALK-rearranged NSCLC [30].

**Figure 2 diagnostics-14-00048-f002:**
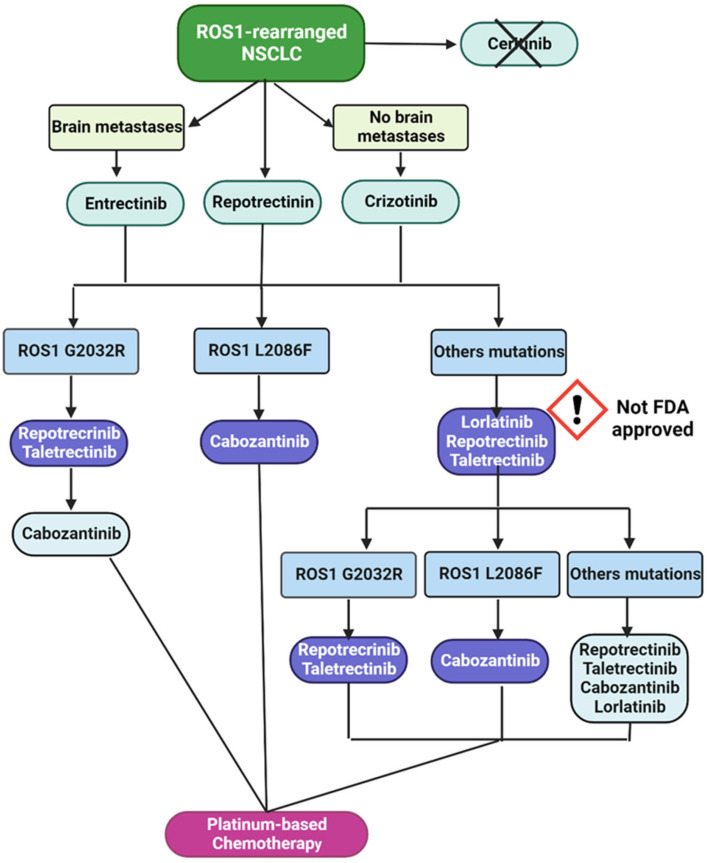
Hypothetical future ROS1-rearranged NSCLC therapeutic algorithm according to emerging secondary resistance mutations [30,66,68].

**Figure 4 diagnostics-14-00048-f004:**
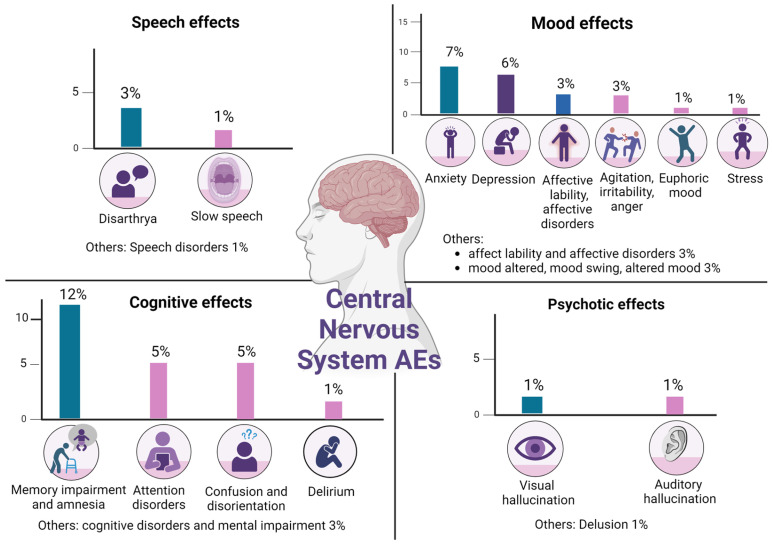
Lorlatinib CNS is the most common AE [25,27].

**Figure 5 diagnostics-14-00048-f005:**
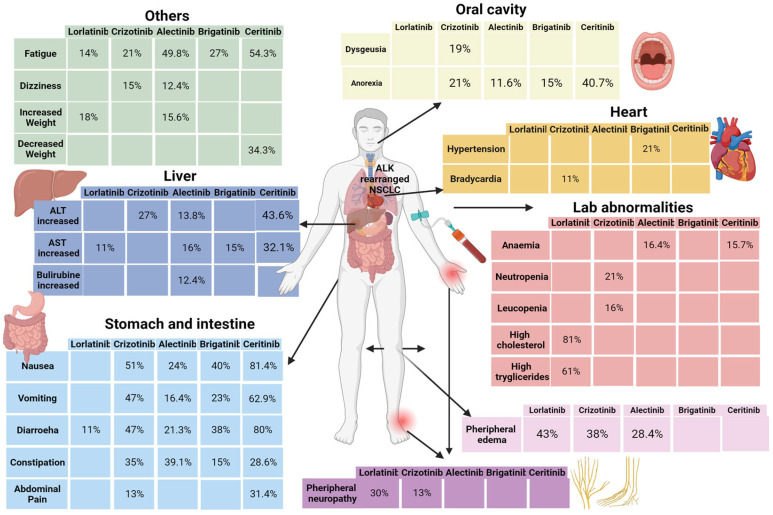
Frequency of systemic AEs was reported in at least 10% of all patients during treatment with different generation ALK TKIs [25,193,194,195,196].

**Table 1 diagnostics-14-00048-t001:** Studies table.

N°	Phase	Patients	Study Description	Trial Arms	ORR %	DoR Months	IC-ORR %	IC-DoR Months	PFS Months	OS Months
1	II	296	LOR vs. CRI in ALK-R NSCLC 1L	LOR	76		66			NR
				CRI	58		20			NR
2	II	296	LOR vs. CRI in ALK-R NSCLC 1L	BM					NR vs. 7.2	
				Not-BM					NR vs. 11	
3	II	228	LOR in NSCLC according to prior therapy and ALK mutational status	CRI	73				NR	
				2G TKI	40				6.9	
				CRI-ALK+	69				12.5	
				CRI-ALK-	74				12.5	
				2G-ALK+	69				7.3	
				2G-ALK-					5.5	
4	II	275	LOR in ALK-R therapy naïve, pretreated with CRI or others TKI or CHT NSCLC	Naïve	90	NR	66.7	NR	NR	
				CRI ± CHT	69.5	NR	87	NR	NR	
				≥1 TKI	47	NR	63	14.5	7.3	
				TKI non-CRI ± CHT	32.1	NR	55.6	NR	5.5	
				≥2 TKI	38.7	NR	53.1	14.5	6.9	
5	III	296	LOR vs. CRI in ALK-R therapy-naïve NSCLC	LOR					NR	
				CRI					9.3	
6	I-II	69	LOR in ROS1-R naïve or pretreated NSCLC	All	41					
				CRI	35	13.8	50	1.4–20.7	8.5	
				naïve	62	25.3	64	1.4–34.7	21	
7	I	54	LOR in ALK-R and ROS1-R naive or pretreated NSCLC	ALK	46	12.4	42		9.6	
				ALK-1 TKI	57					
				ALK- >1 TKI	42					
				ROS1	50	16.6	60		7	
				ROS1-CRI	17					
				ROS1-CHT	33					
8	II	109	LOR after CRI or another ALK TKI ± CRI in ALK-R NSCLC	CRI	70.1	NR	80.6	NR	NR	NR
				TKI not CRI	47.6	11.2	47.6	NR	5.6	NR
9	R	208	LOR in ALK-R pretreated NSCLC	LOR	49	14.9	56	16.7	9.9	32.9
10	R	80	LOR in ROS1-R NSCLC	LOR	45		72		7.1	51.9
11	R	38	LOR in ALK-R naïve or pretreated NSCLC	LOR	44		35		7.3	45
12	R	38	PEM vs. LOR after ALE in ALK-R NSCLC	PEM	45				6.9	16.6
				LOR	44				6.2	17.7
13	R	51	LOR in ROS1-R and ALK-Rpretreated NSCLC	ALK	43.2		62.5			10.2
				ROS	85.7					20
14	R	123	LOR in ALK-R and ROS1-Rpretreated NSCLC	ALK	60		62			89.1
				ROS	62		67			90.3
15	II	16	LOR in ROS1-R NSCLC with brain-only progression on CRI	LOR			87		IC: 38.8 EC: 41.1	
16	R	109	LOR in ALK-Rpretreated NSCLC	LOR	35.7				6.2	NR
17	II	139	LOR in ALK-Rpretreated patients	≥1 TKI 2G	39.6	9.6	56.1	12.4	6.6	20.7
				1 TKI 2G	42.9	6.2	66.7	20.7	5.5	38.5
				≥2 TKI	38.7	9.9	54.2	12.4	6.9	19.2
18	R	10	LOR in N-line after CRI failure in ALK-R NSCLC	LOR	70	9			9	
19	R	51	LOR in 2L or ≥ 3L after ALE in ALK-R NSCLC	Any L	35.7	11.1				
				2L	44.	10.8				
				≥3L	23.5	11.5				
20	R	12	LOR in ALK-R and ROS1-Rpretreated patients	ALK	67%	5	100		6.5	NR
				ROS1	50%					
21	R	95	LOR in ALK-R and ROS1-R NSCLC according to previous treatments	All ALK	33		35		9.3	
				ALK-1 TKI	22				9.3	
				ALK-1 TKI 2G	13				9.2	
				ALK-2 TKI	42				NR	
				ALK- ≥2 TKI	35				11.2	
				ALK- ≥3 TKI	18				6.5	
				ROS1	41		55		11.9	
				ROS1-1 TKI	27				9.7	
				ROS1-1 TKI 2G						
				ROS1-2 TKI	60				11.9	
				ROS1- ≥2 TKI	67				11.9	
				ROS1- ≥3 TKI						

LOR = Lorlatinib; CRI = Crizotinib; ORR = Overall Response Rate; DoR = Duration of Response; IC-ORR = IntraCranial Overall Response Rate; IC-DoR = IntraCranial Duration of Response; PFS = Progression Free Survival; NR = Not Reached; BM = Brain Metastases; 2G = Second Generation; TKI = Tyrosine Kinase Inhibitor; ALK = Anaplastic Lymphoma Kinase; CHT = Chemotherapy; NSCLC = Non Small Cell Lung Cancer; PEM = Pemetrexed; ALE = Alectinib; 1–2–3L = First–Second–Third Line; R = Retrospective [18,25,26,27,37,38,39,40,41,42,43,44,45,46,47,48,49,50,51,52,53].

**Table 2 diagnostics-14-00048-t002:** Single and compound secondary resistance mutations to common ALK inhibitors in vitro and in vivo [42,70,77,96,97,98,99,100,103,104,105,106,107,108,109,110,111,112,113,114,115,116,117,118,119,120].

Mutations	Crizotinib	Ceritinib	Alectinib	Brigatinib	Lorlatinib
WT					
C1156Y					
D1203N					
E1210K					
F1174C					
F1174L					
F1174V					
F1245C					
G1123S					
G1202del					
G1202R					
G1206C					
G1206Y					
G1269A					
G1269S					
I1171N					
I1171S					
I1171T					
L1151tins					
L1196M					
L1198F					
L1152P					
L1152R					
L1256F					
T1151K					
T1151R					
V1180L					
C1156Y + I1171T					
C1156Y + F1174 C/I/V					
C1156Y + L1196M					
C1156Y + L1198F					
C1156Y + D1203N					
C1156Y + S1256F					
C1156Y + G1269A					
D1203N + E1210K					
D1203N + G1123D					
F1174C + G1269A					
G1202R + G1269A					
G1202R + L1196M					
G1202R + F1174L					
G1202R + F1174C					
G1202R + T1151M					
G1202R + F1174L					
G1202R + L1198F					
I1171N + F1174L					
I1171N + F1174I					
SI1171N + L1196M					
I1171N + L1198F					
I1171N + L1198H					
I1171N/T + L1198F					
I1171S + G1269A					
I1171N + L1256F					
I1171N + G1269A					
I1171T + G1269A					
I1171N + D1203N					
I1171T + D1203N					
L1196M + G1269A					
L1196M + D1203N					
L1196M + I1171S					
L1196M + F1174C/L/V					
L1196M + L1179V					
L1196M + L1198F/H					
L1196M + L1256F					
S1206F + G1202R + G1269A					
G1202R + G1269A + L1204V					
G1202R + L1204V + G1269A					
L1196M + G1202R + I1171N					
E1210K + D1203N + G1269A					
G1202R + L1196M + C1156Y					
Legend					
	IC_50_ < 50 nmol/L in all studies or in clinical practice			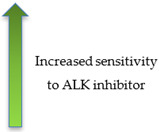	
	IC_50_ < 50 nmol/L in some studies but 50 nmol/L < IC_50_ < 200 nmol/L in other ones.				
	50 nmol/L< IC_50_ < 200 nmol/L in all studies, or evidence of both sensitivity and resistance in clinical practice				
	50 nmol/L < IC_50_ < 200 nmol/L in some studies, but 200 nmol/L< IC_50_ in other ones.			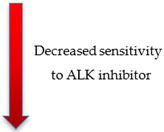	
	IC_50_ > 200 nmol/L in all studies or evidence of resistance in clinical practice				
	IC_50_ not determined				

**Table 3 diagnostics-14-00048-t003:** Ongoing trials [197,198,199,200,201,202].

Trial Identifier	Description	Patients Number	Last Update	Phase	Primary Outcome	Recruitment Status
**NCT05144997**	Lorlatinib safety FU	200	27/03/23	IV	AEs SAEs	Recruiting
**NCT04127110** **(ALKALINE)**	Activity of lorlatinib based on ALK resistance mutations detected in blood in 2G TKI-pretreated ALK+ NSCLC	100	17/02/23	II	PFS	Recruiting
**NCT04111705** **(ORAKLE)**	Lorlatinib in 2L after 1L 2G ALK TKI in advanced ALK+ positive NSCLC	112	19/04/22	II	ORR	Recluiting
**NCT05740943**	Induction lorlatinib for ALK+ Locally advanced NSCLC	48	23/05/23	II	pCR	Recluiting
**NCT04621188** **(ALBATROS)**	Lorlatinib after failure of 1L TKI in advanced ROS1-positive NSCLC	84	10/08/22	II	ORR	Recruiting
**NCT03612154**	Lorlatinib in TKI-naïve and/or CHT-pretreated advanced ROS1 rearranged NSCLC	35	06/04/22	II	ORR	Recruiting

## Data Availability

No new data were created or analyzed in this study. Data sharing is not applicable to this article.

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
