# Peer review of "From Development to Place in Therapy of Lorlatinib for the Treatment of ALK and ROS1 Rearranged Non-Small Cell Lung Cancer (NSCLC)"

_diagnostics, 2023, doi:10.3390/diagnostics14010048_

Round 1

Reviewer 1 Report

Comments and Suggestions for Authors

The current review manuscript titled "From development to place in therapy of Lorlatinib for the treatment of ALK and ROS1 rearranged non-small cell lung cancer (NSCLC)" by Fabbri et al. is a perfectly crafted review that details Lorlatinib for treating NSCLC.

The authors have considered all the aspects, from the mechanism of the drug to clinical trials, when describing Lorlatinib's journey to the clinic. The authors have provided a comprehensive and descriptive Introduction to open up to explore Lorlatinib's role in treating ALK and ROS1-altered NSCLCs. Then, the authors briefly described ALK and ROS1 aberrations leading to NSCLCs. The authors have provided an in-depth review of Lorlatinib from preclinical studies to PK/PD studies. Next, the review highlights the use of Lorlatinib chemotherapy and provides easy-to-understand graphical schemes for both ALK-rearranged and ROS1-mutated NSCLCs.

Further, the table explained different in-depth descriptions of patient studies. The authors have complimented this review by thoroughly reviewing ALK inhibitor resistance. Additionally, focusing on Lorlatinib, the authors have described off-targets and schematically depicted the potentially targetable resistance strategies underlying the necessity of repeating biopsy at progression to detect resistance mutations. The authors have also provided a well-detailed review of the safety profile and clinical management of Lorlatinib chemotherapy and compared the adverse events with other similar therapeutic drugs. In conclusion, the authors have described Lorlatinib very well and successfully put together a comprehensive review describing it as an ALK inhibitor that has the potential to overcome resistance faced by other ALK inhibitors. The authors have also provided reviewed evidence that ROS1 rearranged NSCLCs are efficaciously treated with Lorlatinib treatment as a second-line therapy. This article meets high standards for publication in MDPI's "Diagnostics" in its present form with minor changes.

However, I request that the authors pay attention to providing appropriate references in the introduction section and also change the reference formatting since bold numbers present it until line 120. After that, it is not in bold.

Author Response

Thank you for taking the time to review our manuscript and bringing the issue to our attention. We agree with your comment and have made the necessary revisions. Specifically, we have updated the bibliography citations in the introduction section and removed the bold formatting from the first 120 reference lines to ensure consistency in formatting. 

Reviewer 2 Report

Comments and Suggestions for Authors

Progress which has been observed in oncological treatment in recent years, especially in the field of targeted and biological treatment, increases the survival rate in patient with lung cancer. Authors of this manuscript  aim to explore the role of the third-generation Lorlatinib in the treatment of NSCLCs presenting the ALK and ROS1 aberrations. In this review authors describe detailed information about preclinical development, pharmacodynamic and pharmacokinetics, side effects and clinical utility of lorlatinib.

I find the manuscript as very valuable, graphics are very clear and aesthetic. 

Suggestions:

- table 1 is slightly illegible due to large amount of information, I would suggest to try to make it more clear.

Author Response

Thank you for reviewing the manuscript and bringing this to our attention. We fully agree with your comment and have decided to improve the readability of the table by removing the columns that are related to the TFR data, as they are of minor interest in the context of this review.